# MT2: Towards a Multi-Task Machine Translation Model with Translation-Specific In-Context Learning

**Chunyou Li**[1*], **Mingtong Liu**[2], **Hongxiao Zhang**[1], **Yufeng Chen**[1], **Jinan Xu**[1†], **Ming Zhou**[2]

[1] Beijing Key Lab of Traffic Data Analysis and Mining,
Beijing Jiaotong University, Beijing, China

[2] Beijing Lanzhou Technology Co., Ltd., Beijing, China.

`{chunyouli,hongxiaozhang,chenyf,jaxu}@bjtu.edu.cn`
`{liumingtong,zhouming}@langboat.com`

## Abstract

Sentence-level translation, document-level translation, translation memory, and terminology constrained translation play an important role in machine translation. Most of the previous work uses separate models or methods to solve these tasks, which is not conducive to knowledge transfer of different tasks and increases the complexity of system construction. In this work, we explore the potential of pre-trained language model in machine translation tasks and propose a **M**ulti-**T**ask **M**achine **T**ranslation (MT2) model to integrate these translation tasks. We design a novel translation-specific In-Context Learning (ICL) paradigm for model training, in which all of the translation tasks can be modeled as context-learning tasks that integrate contextual information for performance improvement. Specifically, we propose a retrieval and alignment method to obtain a large scale context-enhancement training data, then we train the model in an in-context learning manner. Furthermore, we adopt two context-dependent training strategies to encourage the model to better understand and utilize contextual information for translation. Extensive experiments on translation memory, terminology constrained translation, document-level translation, and few-shot domain-adaptation tasks demonstrate the superior performance of our model, verifying the effectiveness of our proposed approach.

## 1 Introduction

Translation memory, terminology constrained translation, and document-level machine translation play an important role in machine translation. Most previous studies employ separate models or methods to handle these tasks, such as introducing additional context encoders (Zheng et al., 2020; Xu et al., 2021) and hierarchical attention to modeling

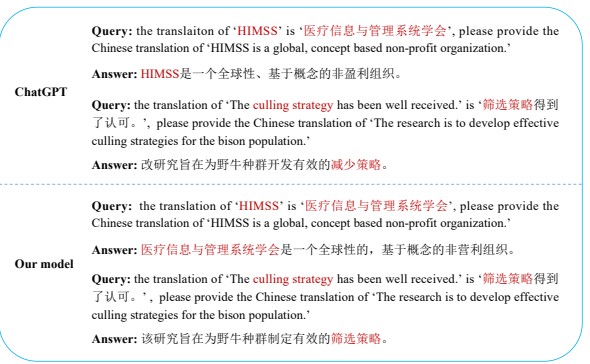

Figure 1: The output of our model and ChatGPT. It shows that our model can effectively use the term and translation memory information for translation while ChatGPT does not perform well.

context information (Tan et al., 2019) in document-level translation task, or changing the model structure (Gu et al., 2018; He et al., 2021, 2019; Khandelwal et al., 2020a) to integrate translation memory sentences. This hampers the transfer of knowledge across diverse tasks and increases to the intricacy of system development.

Recently, large language models (LLMs) have gained significant attention in the field of natural language processing. There are several work that are interested in the application of LLMs in machine translation and have conducted many evaluation work with LLMs, such as GPT3.5 and BLOOM-176B (Scao et al., 2022; Moslem et al., 2023; Peng et al., 2023). These work find that even the currently best-performed LLM, ChatGPT, still lags behind the powerful supervised models in many cases (Zhu et al., 2023; Jiao et al., 2023). Meanwhile, some work has shown that the LLMs can perform better when given appropriate prompts, which shows the in-context learning ability of large models (Ghazvininejad et al., 2023; Peng et al., 2023; He et al., 2023; Brown et al., 2020). These work mainly focus on evaluating and how to stimulate the performance of large language models,

---

[*] Contribution during internship at Beijing Lanzhou Technology Co., Ltd., Beijing, China.

[†] Corresponding author.

however, we focus on training a translation-specific model to enhance in-context translation capability.

In this work, we explore training a unified model to address multiple machine translation tasks. We propose a **M**ulti-**T**ask **M**achine **T**ranslation (MT2) model, which follows the in-context learning paradigm to tackle the context-learning translation tasks that integrate contextual information for performance improvement, such as translation memory, terminology constrained translation, and document-level machine translation et al. We propose a two-stage training method, where the first stage focuses on training the general translation ability, and the second stage aims to improve its context-modeling ability with context-enhancement data and training strategies. Especially, we propose a method based on retrieval and alignment to obtain a large-scale context-enhancement training data, where the retrieved similar sentences and aligned translation pairs are used as contexts for improving translation. Moreover, we adopt two context-dependent training strategies to enhance the context-modeling ability of the model, which encourage the model to better understand and utilize the knowledge transmitted through the context. We give some examples to compare our model with the ChatGPT, and it is shown in Figure 1.

We conduct extensive experiments on multiple translation tasks, including translation memory, terminology constrained translation, document-level translation, and domain adaptation tasks. With the help of in-context learning paradigm, the translation quality and term translation accuracy achieve significant improvement. In addition, the model can perform domain-specific translation directly by concatenating few in-domain examples without updating parameters.

Our contributions can be summarized as follows:

- We propose a multi-task machine translation model with translation-specific in-context learning, which can address multiple translation tasks with the in-context learning paradigm.

- We propose a context-enhancement data construction method based on retrieval and alignment to improve the context learning ability of the model and apply two context-dependent training strategies to promote it to better understand and utilize contextual information.

- We construct extensive experiments on multiple machine translation tasks, and the results demonstrate that our model-7B is comparable to the ChatGPT in in-context learning ability.

## 2 Related Work

### 2.1 Machine Translation Tasks

There are many tasks in the field of machine translation, and most studies have proposed various solutions for different tasks (Liang et al., 2021a,b, 2023). In the translation memory task, the previous work tried to change model structure for integrating translation memory sentence (Gu et al., 2018; He et al., 2021), or introduce additional modules (He et al., 2019; Khandelwal et al., 2020a). Reheman et al. (2023) proposed to treat similar sentences as prompts to the neural machine translation model during the decoding process, which can leverage the knowledge provided by prompts to improve the translation. Recent research showed that LLMs can improve translation performance when prompted by translation memory (Moslem et al., 2023). Typical solutions for terminology constrained translation are based on slot replacement and term injection (Ailem et al., 2021; Anastasopoulos et al., 2021; Zhang et al., 2021). Recent work explored providing phrase-level prompts to the model to improve translation results, which achieves significant gains and demonstrates translation controllability (Sun et al., 2022a; Ghazvininejad et al., 2023). With document-level translation task, some methods focus on changing the architecture of the model, such as introducing additional context encoders (Zheng et al., 2020; Xu et al., 2021), cache-like memory network (Tu et al., 2018), and hierarchical attention (Tan et al., 2019). Sun et al. (2022b) proposes a multi-resolutional training method, which involves multiple levels of sentences and can translate document in an end-to-end manner. There are two common solutions in domain adaptation tasks, which are data centric (Britz et al., 2017) and model centric (Lachaux et al., 2020). More recent methods aim to adapt to a new domain during inference without additional training (Khandelwal et al., 2020b; Zheng et al., 2021; Agrawal et al., 2022; Sun et al., 2022a), where their practice is often to provide some beneficial prompting information.

However, performing each task with separate models may not be conducive to knowledge transfer between different tasks. Our method integrates

these tasks into a unified model with in-context learning paradigm, benefiting knowledge transfer and reducing the intricacy of system development.

## 2.2 In-Context Learning

Recently, in-context learning becomes a new paradigm for natural language processing (NLP), which allows the language models to make predictions only based a few examples in the form of demonstration (Brown et al., 2020; Dong et al., 2022; Moslem et al., 2023). From a principle perspective, the LLMs will learn the pattern hidden in the demonstrations and make correct predictions accordingly. On the other hand, ICL is similar to the decision-making process of human beings, as it involves learning from analogy (Dong et al., 2022). Specifically, given a query input text $q$ and a series of candidate answers $Y = \{y_1, ..., y_n\}$, a well-trained model $\mathcal{M}$ will give the answer conditional on $n$ demonstrations $C = \{(q_1, y_1), ..., (q_n, y_n)\}$, where $(q_i, y_i)$ is a query-answer pair and generally written in task-related natural language templates.

ICL has multiple attractive advantages, the main one is that it can directly perform predictions based on the pre-trained models without updating parameters. This could greatly reduce the computation costs for adapting new tasks (Chen et al., 2021; Min et al., 2022). In addition, it can easily integrate human knowledge into language models by modifying templates and demonstrations (Lu et al., 2022; Wei et al., 2022; He et al., 2023). There are recent works that leverage in-context learning paradigm to solve specific tasks such as information extraction, sentiment analysis (Yoo et al., 2021; Liu et al., 2022; Xu et al., 2023), and machine translation (Moslem et al., 2023; Zhu et al., 2023).

Our work is based on machine translation with in-context learning paradigm, we explore training a unified model to solve multiple translation tasks. Most previous work mainly focus on how to apply on LLMs, while we concentrate on how to train and enhance in-context learning ability of the model for translation task.

## 3 Methodology

### 3.1 Translation-Specific-In Context Learning

In this work, we extend the traditional machine translation training task. When predicting the current token, our model not only considers the source sentence and a part of translated tokens, but also rich context information such as term informa-

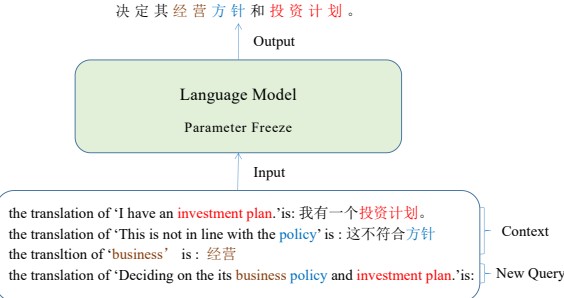

Figure 2: The In-context learning paradigm for machine translation, where the context can be the translation memory sentences, terminologies and preceding translated sentence pairs in the document.

tion, which affects the current prediction with attention mechanism. Like traditional translation task (Vaswani et al., 2017), we adopt the cross-entropy loss function as follow:

$$\mathcal{L}(\theta) = -\sum_{i=1}^{|y|} log p(y_i|y_{<i}; context; \theta),$$

where $\theta$ is the model parameters, $y_i$ and $|y|$ are the $i$-th token in the input sentence $y$ and length of $y$, respectively. $context$ represents the context information.

In the translation memory task and terminology constrained translation task, $context$ represents the similar parallel sentences pairs retrieved and terminology constrained pairs, respectively. For the document-level translation task, $context$ is the preceding source sentences and their corresponding translations.

Specifically, take the translation memory task as an example, given a parallel translation pair $(x, y)$ and a set of retrieved similar sentence pairs $C = \{(x_1^m, y_1^m), ..., (x_n^m, y_n^m)\}$, we wrap each pair $(x_i^m, y_i^m)$ with an instruction template and then concatenate them in front of $x$ as context-enhancement examples. The examples of training data is illustrated in the right of Figure 3.

As illustrated in Figure 2, with the help of the in-context learning paradigm, the model can obtain and utilize knowledge transmitted through context to guide the translation process, so as to make the translation more accurate and improve the quality of translation.

### 3.2 In-Context Learning Data Construction

To train the model with the in-context learning paradigm, we propose a method based on retrieval

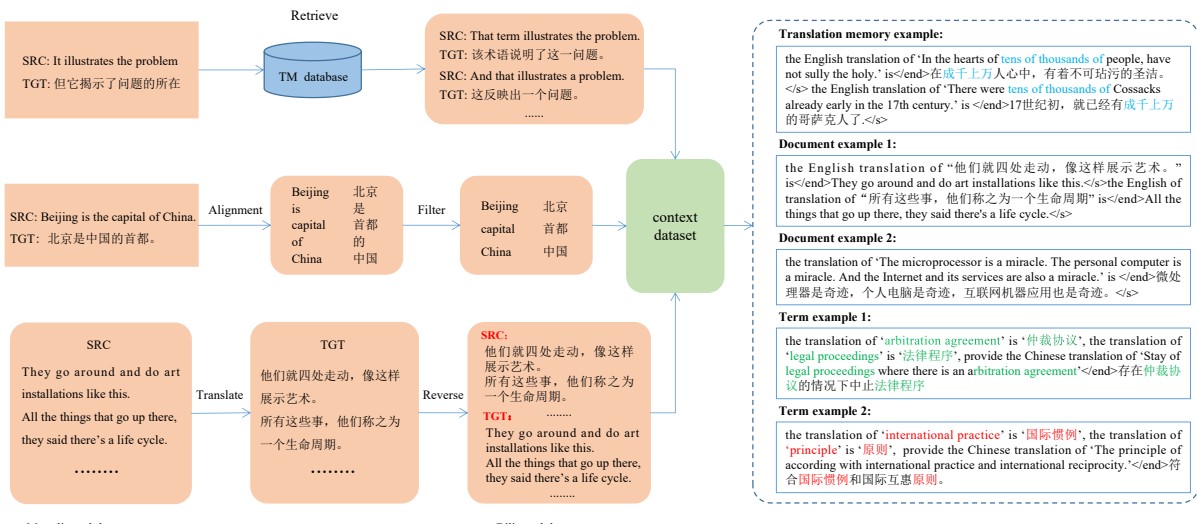

Figure 3: Overview of data construction. The left is methods for obtaining context data, and the right is training data samples. 'Reverse' means to reverse the 'SRC' and 'TGT'. '</end>' is the defined instruction end symbol.

and alignment to construct context-enhancement data. The overview of data construction flow is shown on the left of Figure 3.

**Sentence-level Context Retrieval.** For each source sentence, we assign a similarity score between it and each sample in the translation memory database, where the similarity score is calculated according to the BM25 algorithm (Robertson et al., 2009), and the calculation formula is as follows:

$$score = \sum_{i=1}^{n} w(q_i) . \frac{f(q_i) \cdot (k_1 + 1)}{f(q_i) + k_1 \cdot (1 - b + b \cdot \frac{|D|}{avgdl})}$$

where $q_i$ is the $i$-th word in the retrieval sentence, $D$ is an example in the translation memory database. $w(\cdot)$ denotes inverse document frequency (IDF), and $f(q_i)$ represents the frequency of word $q_i$ in $D$, $k_1$ and $b$ are two adjustable parameters. $|D|$ and $avgdl$ denotes the length of $D$ and average length, respectively. We then rank the similarity scores from largest to smallest and select the top 20 samples as candidate translation memory sentences.

**Phrase-level Context Extraction.** Limited by the scarcity of terminology translation data, we generalize it to phrase-level constrains to construct context-enhancement data. As shown in Figure 3, for each parallel translation pair, we first get the phrase alignment dictionary and then we filter the phrase alignment dictionary according to part of speech, in which we mainly keep noun phrases and adjectives. In this case, we can construct large-scale context-enhanced training data, which is conducive to the training of ICL strategy and makes up

for the lack of data in term-constrained translation task.

**Document-level Context Construction.** We expect the system can model the context of document-level data. However, the parallel document data is scarce, so we try to translate the monolingual document data into the target language with a powerful translation system to obtain document-level training corpus. Specifically, given an English document corpus, we employ a translation model to translation it into Chinese. When used for training, the Chinese is used as the source side and the English is used as target side.

Next, we construct the document-level context-enhancement data in two ways.

- The document data is first split into sentence level. Then like the document example 1 in Figure 3, the source and corresponding translation are concatenated in sentence order.

- To enhance document-level context modeling capability, inspired by (Sun et al., 2022b), we collect $n$ sentences as a segment on the basis of following the sentence order, like document example 2 in Figure 3, where $n$ is a random number.

### 3.3 Context-Learning Enhancement with Two-Stage Training

In order to train the model to better utilize context, we propose to train the model in two stages, namely translation pre-training and in-context learning

training. The first stage mainly cultivates the general translation ability of the model. We use a large number of parallel sentence pairs for training on the basis of the pre-trained language model to enhance its translation ability.

The second stage of training focuses on improving the in-context modeling ability of the model. To this end, the model is trained with the context-enhancement data described above. In addition, to maintain the general translation ability in the model, we also mix some general parallel data.

Furthermore, we adopt two context-dependent training strategies to enhance the contextual learning ability of the model. The first is mask strategy, given a translation memory training sample $< (x_1, y_1, I_1) \ominus (x_2, y_2, I_2) \ominus ... \ominus (x_n, y_n, I_n) >$, there is a 50% probability to perform the following processing: Iterate over each token in sentence $x_n$, and if it has appeared in a context sentence $x_{<n}$, it will be masked with a 25% probability and this token in the context will also be masked with a small probability. Where $\ominus$ denotes concatenation operation, $I_i$ is the instruction template, and $x_i$ and $y_i$ denote the source and target sentence, respectively. In this manner, the model will pay more attention in context when predicting the next token.

In order to enhance the robustness of the model, we propose the second training strategy . Specifically, we preprocess the alignment dictionary to get a new dictionary where each phrase has multiple meanings, that is, polysemy. In practice, each phrase maintains the correct translation with a 50% ratio, and the rest case we will add noise. If the i-th phrase is chosen to be corrupted, it may be replaced with another meaning of the word 30% of the time or an unrelated translation 20% of the time. In this case, it can improve contextual adaptation of terms.

## 4 Experiments

### 4.1 Datasets

For the training data, we collect about 100 million parallel data and about 100 million back-translation data, and we take about 40 million parallel data as the translation memory database. In the second training stage, we supplement about 1.7 million document-level translation corpus.

In the translation memory and domain-adaptation tasks, we evaluate our model in Education, Subtitle, Thesis, and Spoken domains from UM-Corpus (Tian et al., 2014). And we randomly select 1000 sentences for each domain as the test

| | Domain | Education | Subtitle | Thesis | Spoken |
|---|---|---|---|---|---|
| En-Zh | w/o TM | 36.66 | 22.61 | 39.78 | 34.52 |
| En-Zh | w/ TM | **38.83** | **25.22** | **41.85** | **36.38** |
| Zh-En | w/o TM | 25.02 | 20.64 | 17.79 | 24.56 |
| Zh-En | w/ TM | **28.14** | **22.19** | **21.80** | **26.64** |

Table 1: Experiments results on the translation memory and domain-adaptation tasks.

dataset, whereas the remaining data are utilized as the translation memory database. In the document-level translation task, we combine tst2010-2013 from IWSLT2015 as the test dataset in the Zh-En direction. We also select a representative document-level novel translation in the En-Zh direction. Previous work demonstrated the importance of contextual information in ensuring consistency and coherence in novel translation (Wang et al., 2023; Karpinska and Iyyer, 2023). We manually constructed a high-quality test dataset of 1000 sentences. In the terminology constrained translation task, we construct test dataset for Aviation, Law, and Entertainment domains based on a terminology dictionary, and each test dataset contains 2000 sentences. Specifically, we use a 30w collated terminology dictionary. The evaluation metric is Sacre-BLEU[1]. We also the report COMET (Rei et al., 2022) scores in Appendix A.

### 4.2 Implementation Details

For the model setting, we choose the BLOOM-7b1 as the base model, which is an decoder-only transformer language trained by Bigscience (Scao et al., 2022). All our methods are implemented with Huggingface and Deepspeed-Chat[2]. We apply zero-stage-3 (Rajbhandari et al., 2020) and gradient_checkpointing strategy to train the model with 32 A100 GPUs. We set the batch_size and gradient_accumulation_steps hyperparameters to 5, 32, respectively. The learning rate for the first stage is 1e-4 and the learning rate for second stage is 5e-5. And the parameters are optimized by using Adam optimizer with $\beta_1 = 0.9, \beta_2 = 0.95$ and $\epsilon = 10^{-8}$. We obtain the phrase alignment dictionary with fast_align (Dyer et al., 2013).

---

[1] https://github.com/mjpost/sacrebleu
[2] https://github.com/microsoft/DeepSpeedExamples

| Domain Method | Aviation | | Law | | Entertainment | |
|---|---|---|---|---|---|---|
| | BLEU | Accuracy | BLEU | Accuracy | BLEU | Accuracy |
| *w/o* Term | 41.61 | 81.44% | 53.62 | 84.37% | 30.77 | 78.57% |
| *w* Term | **44.93** | **92.27%** | **56.13** | **92.28%** | **31.81** | **87.07%** |

Table 2: The performance of model on the terminology constrained translation task in En-Zh direction. Accuracy represents the rate of the target phrase appears in the translation output.

| Task Model | Translation memory task | | | Terminology constrained translation task | | |
|---|---|---|---|---|---|---|
| | *w/o* ICL | *w/* ICL | $\Delta$ | *w/o* ICL | *w/* ICL | $\Delta$ |
| ChatGPT | 32.26 | 36.41 | 4.15 | 48.61/85.03 | 51.33/94.94 | 2.72/**9.91** |
| Ours | 36.03 | 40.27 | **4.24** | 51.69/83.97 | 56.18/93.67 | **4.49**/9.70 |

Table 3: The comparison between the ChatGPT and our model in translation memory and terminology constrained tasks. We also test the accuracy of term translation in terminology constrained translation task. Our model achieves greater growth than ChatGPT with the in-context learning paradigm.

## 4.3   Main Results

**Translation Memory and Domain Adaptation.** In this experiment, we evaluate how well the model performs domain-adaptation task directly without parameter updates. To be more convincing, we discard the translation memory sentence which is the same as the sentence to be translated and the results are given in Table 1. When the similar in-domain sentences retrieved are provided, the performance increases significantly, with average BLEU score improvements of 2.18 and 2.69 points in En-Zh and Zh-En directions, respectively. The results show that the model has the ability to capture the knowledge contained in the context and apply it to improve translations, and the higher the sentence similarity, the more useful information is provided. **Terminology Constrained Translation.** To evaluate the performance of the model in terminology task, we conduct experiments in three datasets and the results are reported in Table 2. With the help of terminology constrained information, our model can significantly improve the accuracy of terminology constrained translation in all datasets without additional training. There was also an improvement in the overall translation quality evaluated by BLEU scores,with approximately 3 points. These experiments show that our model can effectively utilize the knowledge transmitted from the contexts and apply it to translation processing. **Document Translation.** To show the context-learning performance of the model in document-level translation, we conduct experiments with and

| Direction | System | ICL | BLEU |
|---|---|---|---|
| Zh-En | Base | No | 30.72 |
| | Base | Yes | 29.38 |
| | MT2 | No | 30.50 |
| | MT2 | Yes | **31.33** |
| En-Zh | Base | No | 28.54 |
| | Base | Yes | 29.08 |
| | MT2 | No | 28.78 |
| | MT2 | Yes | **29.50** |

Table 4: Experimental results of document-level translation in Zh-En and En-Zh directions. 'Base' and 'MT2' represent the models trained from the translation pre-training and in-context learning stage, respectively.

without the in-context learning paradigm and the results is shown in Table 4. The value of *ICL* is *No* indicates that no contextual information is provided, that is, traditional sentence-level translation. Instead, *Yes* means that when translating the sentence, we will concatenate some preceding source sentences and corresponding translations. In line with the process of human document translation, our method takes into account the historical translation information, ensuring better consistency between the translated segments. From the table, the MT2 performs better than the base model, and we can find that the model can effectively improve the overall translation quality with the in-context learn-

| |
|---|
| I: give the English translation of '我父母于1960年结婚。' |
| T:My parents were married in 1960. |
| C: the translation of '结婚' is 'get married' |
| T: My parents got married in 1960. |

| |
|---|
| I: give the Chinese translation of 'It is to develop effective culling strategies for the bison population.' |
| T: 它旨在发展有效之野牛族群扑杀策略。 |
| C: the Chinese translation of 'The culling strategy has been well received.'is'筛选策略得到了认可。' |
| T: 它旨在为野牛种群制定有效的筛选策略。 |

| |
|---|
| I: give the English translation of '李四比较年轻，工作经验也不足，学历又不高，但是不论做啥事情，他都认真负责，所以，领导非常器重他。' |
| T: Li Si is younger, work experience is also insufficient, education is not high, but no matter do what thing, he is conscientious and responsible, so, the leader thinks highly of him very much. |
| D: If we translate this paragraph with in-context learning paradigm, the translation is as follows: |
| T: Li Si is younger and has less work experience and less education. But no matter what he does, he is serious and responsible. So, the leader thinks highly of him. |

Table 5: Some cases showing different outputs of the model with and without context. Where I, T, C, D represent *Instruction*, *Translation*, *Context* and *Description*, respectively. The third example shows subject consistency with in-context learning paradigm.

ing paradigm. In addition, as shown in Table 5, our model can improve the translation quality in consistency and coherence.

### 4.4 Compared With ChatGPT

In order to explore the performance of our model and ChatGPT in solving translation tasks with in-context learning paradigm, we select 50 examples each for terminology constrained translation and translation memory tasks. We obtain the translation results of ChatGPT through web version[3]. We use two prompt templates and record the higher scores for ChatGPT, *<src>=<tgt>* and *the translation of <src> is <tgt>*, where the *<src>* and *<tgt>* denote the placeholder for source and target sentence respectively. The results are given in Table 3. Due to different training data and model sizes, it is difficult to directly compare with ChatGPT. Results show that our model achieves better translation performance with supervised training data, which is consistent with previous studies (Zhu et al., 2023). In this work, we mainly focus on the performance improvement of the model with and without the in-context learning paradigm, which is the value of $\Delta$ in the table. The results show that our model generally perform better than ChatGPT in terms of performance increase, except for term accuracy. Although our model is small in scale, it can achieve results comparable to ChatGPT after training with

our proposed method for translation task.

### 4.5 Case Study

We give some cases of the output of model with and without providing context in Table 5. We can see that the prompt context can affect the translation results, which indicates that the model has the ability to obtain the information conveyed by the context prompts and apply them to translation process. An interesting example is example 1, where the tense of the information we provide is present simple, and the translation results given by the model apply the correct past tense.

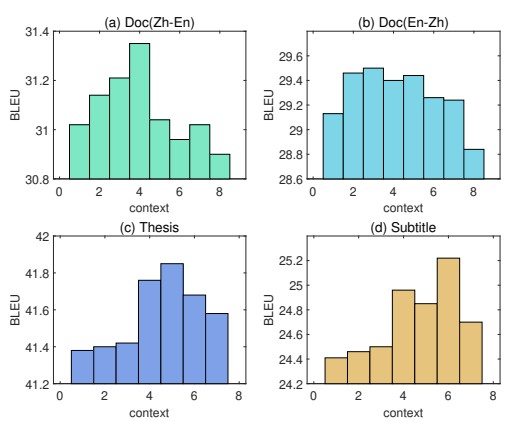

Figure 4: The effect of the number of context.

[3]https://chat.openai.com/

| Domain | Education | Subtitle | Thesis | Spoken |
|---|---|---|---|---|
| farthest | 37.97 | 24.02 | 41.21 | 34.88 |
| middle | 38.06 | 24.04 | 41.29 | 34.93 |
| nearest | **38.64** | **24.22** | **41.42** | **35.71** |

Table 6: Results on the impact of similar sentences order.

| Domain | Education | Subtitle | Thesis | Spoken |
|---|---|---|---|---|
| *w/o* mask | **39.02** | 24.87 | 41.60 | 35.92 |
| *w/* mask | 38.83 | **25.22** | **41.85** | **36.38** |

Table 7: Experiments on the impact of mask strategy.

# 5 Analysis

## 5.1 Effect of the Number of Context

In this subsection, we investigate the effect of the number of concatenated contexts on the translation results. The results is in Figure 4. Overall, the performance improves initially as the number of contexts increases, but as we continue to add more contexts, the scores start to drop. We speculate that as the number of contexts increases, noise will be introduced, resulting in a decrease in translation quality. In the translation memory task, concatenating more contexts means that the retrieved sentences get lower similarity scores and the correlation between texts with longer spans will also decrease in document-level translation, leading to the introduction of noise which degrades the translation quality.

## 5.2 Effect of the Order of Context

It is obvious that the translation results will be affected by the similarity score of the retrieved sentences. Besides, we are interested in evaluating the effect of the concatenation order of similar sentences on the translation results. Hence, we take top three similar sentences retrieved as a baseline to test the influence of the most similar sentence being farthest, middle, and nearest to the sentence to be translated, respectively. And the experiments results are shown in Table 6. In line with the intuitive idea, when the most similar sentence is closer to the sentence to be translated, the performance improvement is more obvious, indicating that the model is more sensitive to closer context information and pays more attention to it.

Figure 5: The comparison results, where the top is the output without the second context-enhancement strategy and the bottom is the output with the strategy. The bottom results are more robust than the one above.

| Method | Aviation | Law | Entertainment |
|---|---|---|---|
| No | 41.38 | 52.45 | 29.73 |
| Yes | **43.66** | **54.98** | **31.50** |

Table 8: The comparison results with and without the second strategy

## 5.3 Impact of Context-Enhancement Strategy

In this subsection, we conduct ablation study to explore the benefits of these two strategies. From the Table 7, we can observe that the model with mask strategy performs better in all datasets except Education, with maximum BLEU score improvements of 0.35, 0.25, 0.46 in Subtitle, Thesis and Spoken datasets, respectively. Therefore, we suggest that with the help of this strategy, the model can effectively pay attention to the useful information contained in the context and utilize it to improve the translation quality.

The reason we introduce the second strategy is to allow the model to adapt to the noise in the context without breaking the semantics of the sentence itself. In order to verify the effectiveness of the method, we introduced some noise into the test set, and the experimental results are shown in Table 8. We also provide a few examples to evaluate the performance of the model in two scenarios in the Figure 5. We can observe that when we give a correct translation pairs in the context, the model can use this information to improve the translation results. However, when we give an incorrect term translation, the model will discard this prompt information as it destroys the semantics of the sentence itself. That shows the performance of the model after adopting the second strategy is more robust and it can reduce the impact of noise in the

context when translating.

# 6 Conclusion

In this work, we propose a multi-task machine translation model with translation-specific in-context learning, which can address multiple context-learning type translation tasks with the in-context learning paradigm. We propose a method based retrieval and alignment to construct the context-enhancement data and then train the model with in-context learning paradigm in two stages. In addition, we also adopt two context-dependent training strategies to make model better understand and utilize contextual information for performance improvement. Finally, we conduct experiments on terminology constrained translation, document-level translation, translation memory and few-shot domain-adaptation tasks to evaluate the effectiveness of our approach.

We will extend the model to a multilingual translation system and explore instruction transfer learning in the future.

## Limitations

Here we discuss the limitations of our work. First, for the convenience of data processing and training, we set the maximum length of input text at training time, which may cause the model to perform poorly when dealing with too long input text. Secondly, the noise contained in the context has a negative impact on translation quality, and we adopted some strategies to enhance the robustness of the model. However, it is still worth exploring how to avoid or alleviate the impact of noise on translation quality. We will conduct further research on the above issues in our future work.

## Acknowledgement

The research work descried in this paper has been supported by the National Key R&D Program of China (2020AAA0108001) and the National Nature Science Foundation of China (No. 61976015, 61976016, 61876198 and 61370130). The authors would like to thank the anonymous reviewers for their valuable comments and suggestions to improve this paper.

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

# A Appendix

| Domain | | Education | Subtitle | Thesis | Spoken |
|---|---|---|---|---|---|
| En-Zh | *w/o* TM | 55.3 | 17.2 | 64.4 | 52.4 |
| | *w/* TM | **57.3** | **21.6** | **65.4** | **53.4** |
| Zh-En | *w/o* TM | 37.4 | 6.40 | 15.6 | 36.2 |
| | *w/* TM | **41.7** | **7.31** | **19.1** | **36.5** |

Table 9: The COMET scores on the translation memory and domain-adaptation tasks.

| Direction | System | ICL | COMET↑ |
|---|---|---|---|
| Zh-En | MT2 | No | 47.2 |
| | MT2 | Yes | **48.1** |
| En-Zh | MT2 | No | 35.7 |
| | MT2 | Yes | **38.0** |

Table 10: The COMET scores of document-level translation in Zh-En and En-Zh directions.

| Domain | Aviation | Law | Entertainment |
|---|---|---|---|
| *w/o* Term | 66.1 | 89.4 | 56.7 |
| *w* Term | **70.5** | **93.0** | **57.9** |

Table 11: The COMET scores of terminology constrained translation task in En-Zh direction.