# OpenReview forum: "MT2: Towards a Multi-Task Machine Translation Model with Translation-Specific In-Context Learning"
_EMNLP/2023/Conference — EMNLP 2023 Main_

### Official Review · Reviewer_qFRK · 2023-08-01

**Soundness:** 4

**Excitement:**

4: Strong: This paper deepens the understanding of some phenomenon or lowers the barriers to an existing research direction.

**Paper Topic And Main Contributions:**

This work proposes a multi-task translation model MT2 with in-context learning, which integrates translation memory, terminology constrained translation, document-level translation, and few-shot domain adaptation tasks. The model is first pre-trained on a huge parallel corpus and then trained on an in-context learning dataset constructed by retrieval and alignment. Extensive experiments show MT2 achieves significant improvement over the base model and ChatGPT.

**Questions For The Authors:**

When constructing sentence-level context, where does the source sentence mentioned in L250 comes from?

**Reasons To Accept:**

* An integrated translation model is useful for real applications.
* The improvement of empirical results is significant compared with the base model and ChatGPT.
* According to the shown examples, the proposed training strategy effectively enhances the translation quality.


**Reasons To Reject:**

* This model is bilingual, not multilingual. It is unclear its performance when generalizing to other languages.
* While the results are impressive, training such a model is expensive (100 million data for one language). Though this may not be a problem for an off-the-shelf model, I am wondering if such amount of data is necessary. If not, MT2 can conveniently expand to other foundation models.


**Reproducibility:**

4: Could mostly reproduce the results, but there may be some variation because of sample variance or minor variations in their interpretation of the protocol or method.

**Reviewer Confidence:**

3: Pretty sure, but there's a chance I missed something. Although I have a good feel for this area in general, I did not carefully check the paper's details, e.g., the math, experimental design, or novelty.

---

> ### Author Rebuttal · Authors · 2023-08-29
>
> Dear Reviewer qFRK:
>
> We sincerely thank you for the time and effort you have dedicated to reviewing our paper. Your valuable comments and constructive suggestions have been instrumental in improving the quality of our work. Below, we reply to the comments raised by the reviewer point-by-point.
>
> **Q1: This model is bilingual, not multilingual.**
>
> A1: As with previous work, in order to verify the effectiveness of the method, we conducted experiments on English-Chinese bidirectional data. However, we can easily extend it to multiple language translation, because BLOOM is a multilingual pre-trained language model. In the future, as you mentioned, we will extend it to a multilingual translation model.
>
>
> **Q2: The performance in the small-scale datasets.**
>
> A2: With the in-context learning paradigm, there is also improvement when there is less data. Under the same data, the gain brought with the in-context learning paradigm is more. When we did the preliminary experiment, we conducted it on a relatively small-scale data, and the final effect tends to be roughly the same on each task. Specifically, we use about 10 million data, and with the in-context learning paradigm, the BLEU score could be increased by 1-2 points. Therefore, our method can also improve the performance of model on small-scale datasets. Thanks for your suggestion, we will add these conclusions in the future to better illustrate our work.
>
>
> **Q3: Where does the source sentence mentioned in L250 comes from?**
>
> A3: There may be ambiguity in my expression, the “source sentence” here refers to the source sentence in the parallel sentence pair. We will correct this description, thanks for the suggestion.\
> \
> \
> Thanking you again for your generosity with your valuable time invested in improving our paper.
>
> Yours sincerely,
>
> All author.

---

### Official Review · Reviewer_7odN · 2023-08-05

**Soundness:** 3

**Excitement:**

3: Ambivalent: It has merits (e.g., it reports state-of-the-art results, the idea is nice), but there are key weaknesses (e.g., it describes incremental work), and it can significantly benefit from another round of revision. However, I won't object to accepting it if my co-reviewers champion it.

**Missing References:**

Min et al. MetaICL: Learning to Learn In Context. NAACL 2022. https://aclanthology.org/2022.naacl-main.201/

Chen et al. Meta-learning via Language Model In-context Tuning. ACL 2022. https://aclanthology.org/2022.acl-long.53/

**Paper Topic And Main Contributions:**

This paper presents an approach (MT2) to train a model that handles multiple in-context translation tasks (e.g. terminology constrained translation, document-level translation, translation memory). A pre-trained language model is first fine-tuned on parallel sentences to improve general translation ability. It is then further finetuned on a dataset, created based on retrieval and alignment, containing in-context prompts for multiple specific translation tasks. During this second stage, some tokens are masked to encourage the model to rely on the context more. Some phrases are also replaced to improve model robustness.

**Questions For The Authors:**

A. In section 3.3, how do you ensure that each phrase has multiple meanings?

B. What is the performance of BLOOM-7b1 if you don't finetune the model, but use the same prompts for inference?

C. Do you have quantitative results for the second context-enhancement strategy?

**Reasons To Accept:**

A unified model can handle multiple translation tasks, and the additional context clearly improves results.

The paper includes many comparative examples that help understand the behavior and benefits of the approach.

**Reasons To Reject:**

The importance of multi-task training is not clearly demonstrated. The MT2 model should be compared with models for which the 2nd stage only contains examples for 1 task (plus general parallel data).

Important related work (see "Missing References") is missed, especially in terms of training a language/translation model on in-context learning examples (i.e. learning to learn in-context).

**Reproducibility:**

3: Could reproduce the results with some difficulty. The settings of parameters are underspecified or subjectively determined; the training/evaluation data are not widely available.

**Reviewer Confidence:**

4: Quite sure. I tried to check the important points carefully. It's unlikely, though conceivable, that I missed something that should affect my ratings.

**Typos Grammar Style And Presentation Improvements:**

L90: we -> We

BM25: add citation

Give a name to the second context-enhancement strategy.

---

> ### Author Rebuttal · Authors · 2023-08-29
>
> Dear Reviewer 7odN：
>
> We sincerely thank you for the time and effort you have dedicated to reviewing our paper. Your valuable comments and constructive suggestions have been instrumental in improving the quality of our work. Below, we reply to the comments raised by the reviewer point-by-point.
>
>
> **Q1: The importance of multi-task training is not clearly demonstrated.**
>
> A1: Our main contribution is to unify multiple tasks into a unified model through the in-context learning paradigm, and propose several solutions to enhance in-context learning. The traditional approach is to model these tasks separately, and some tasks are limited by the limited training data, such as terminology constrained translation task. Our proposed two methods for constructing training data and multi-task training can alleviate the problem of task data sparsity. The importance of multi-task learning is not our main concern,we did not analyze it in detail. Previous work demonstrates that multi-task learning benefits the learning of machine translation models, such as [R1] and [R2]. We followed the successful experience and ideas of previous work. But a major difference is that we propose a method based on the in-context-learning paradigm, which integrates traditional machine translation methods into a unified model, thus different tasks can be jointly trained to improve the effect.
>
> **Q2: Missing References**
>
> A2: Thank you for your reminder, we will add corresponding references [R3], [R4]  and [R7] in later editions.
>
> **Q3: Each phrase has multiple meaning.**
>
> A3: Our motivation is to enhance the robustness of the model. We obtain alignment dictionaries with an alignment tool from a large amount of parallel data in advance, each segment can get multiple meanings with high probability. And it doesn't matter even if it doesn't have multiple meanings, we will keep the original correct interpretation and add random noise when constructing the training data. Thank you for raising this question, we will add corresponding details later.
>
> **Q4: The difference in performance between the bloom-7b1 model and the model after finetuning.**
>
> A4: We evaluated the origin bloom-7b1 model, and the results are not good. The BLEU score is basically below 5, and it is prone to translation off-target and repeated generation problems. The [R5] paper also reached the same conclusion.. Typically, larger models have better translation results, however evan a large model of 176B is still weaker than supervised training translation models (R[6]). Thank you for your suggestion, we will add the corresponding experimental results in the appendix later.
>
>
> **Q5: A quantitative results for the second context-enhancement strategy.**
>
> A5: Thank you for your suggestion. In our experiment, in order to enhance the robustness of the model, we adopted the second strategy. The manual evaluation demonstrated improvements, and we conducted case studies, but we apologize for not including the corresponding statistical analysis process in the article.  We adopt your suggestion and give a quantitative analysis experiment, where we introduce some noise in the terminology translation task. The results is (Aviation: 41.38 vs. 43.66;  Law: 52.45 vs. 54.98;  Entertainment: 29.73 vs. 31.50;). The experimental results show that the model with the second training strategy is more robust. We will include corresponding analysis in future version for better understanding by readers. Thanks again for your suggestion.\
> \
> \
> Thanking you again for your generosity with your valuable time invested in improving our paper.
>
> Yours sincerely,
>
> All author.\
> \
> \
> **References**
>
> [R1] Zhou S, Zeng X, Zhou Y, et al. Improving robustness of neural machine translation with multi-task learning[C]//Proceedings of the Fourth Conference on Machine Translation (Volume 2: Shared Task Papers, Day 1). 2019: 565-571.
>
> [R2] Luong M T, Le Q V, Sutskever I, et al. Multi-task sequence to sequence learning[J]. arXiv preprint arXiv:1511.06114, 2015.
>
> [R3] Min S, Lewis M, Zettlemoyer L, et al. Metaicl: Learning to learn in context[J]. arXiv preprint arXiv:2110.15943, 2021.
>
> [R4] Chen Y, Zhong R, Zha S, et al. Meta-learning via language model in-context tuning[J]. arXiv preprint arXiv:2110.07814, 2021.
>
> [R5] Bawden R, Yvon F. Investigating the translation performance of a large multilingual language model: the case of bloom[J]. arXiv preprint arXiv:2303.01911, 2023.
>
> [R6] Jiao W, Wang W, Huang J, et al. Is ChatGPT a good translator? A preliminary study[J]. arXiv preprint arXiv:2301.08745, 2023.
>
> [R7] Robertson S, Zaragoza H. The probabilistic relevance framework: BM25 and beyond[J]. Foundations and Trends® in Information Retrieval, 2009, 3(4): 333-389.

---

### Official Review · Reviewer_1ZMP · 2023-08-12

**Soundness:** 4

**Excitement:**

4: Strong: This paper deepens the understanding of some phenomenon or lowers the barriers to an existing research direction.

**Paper Topic And Main Contributions:**

The paper presents a novel Multi-Task Machine Translation (MT2) model that effectively incorporates various translation tasks, including sentence-level translation, document-level translation, translation memory, and terminology constrained translation. The model introduces an innovative In-Context Learning (ICL) paradigm, treating all translation tasks as context-learning tasks. Additionally, the authors devise a retrieval and alignment technique for acquiring extensive context-enhancement training data and implement two context-dependent training strategies to enhance the model's comprehension and utilization of contextual information. Experimental results demonstrate the model's excellent performance across multiple machine translation tasks.

**Questions For The Authors:**

What is the difference in performance between the non-finetuned bloom-7b1 model and the model after finetuning?

**Reasons To Accept:**

1. The authors proposed an innovative In-Context Learning (ICL) paradigm, which effectively utilizes information from the historical translation database to enhance translation performance.

2. The authors conducted extensive experiments to verify the effectiveness of the Translation-Specific In Context Learning method for translation tasks, especially conditional machine translation tasks.

3. The authors provided a reproducible data construction method, which allows for the quick and easy collection of a large amount of data, facilitating replication and further exploration by future researchers.

**Reasons To Reject:**

1. In the era of Large Language Models, traditional evaluation metrics such as BLEU and ROUGE cannot accurately judge the translation quality. Therefore, it is best to combine manual evaluation, training-based automated metrics, or incorporate GPT-4 for scoring, especially when the test dataset is limited and small. This would provide a more credible assessment.

2. The paper relies heavily on case analysis to support its conclusions, which is not sufficiently persuasive. For example, the statement in the paper, "In addition, as shown in Table 5, our model can improve the translation quality in consistency and coherence."

3. The paper should include quantitative analysis results for the second training strategy, rather than just relying on a case analysis. This is necessary to validate the effectiveness and necessity of the strategy.

4. It would be preferable to disclose more details about the data evaluation, such as the test dataset. This would facilitate replication and further exploration by other researchers.

**Reproducibility:**

4: Could mostly reproduce the results, but there may be some variation because of sample variance or minor variations in their interpretation of the protocol or method.

**Reviewer Confidence:**

4: Quite sure. I tried to check the important points carefully. It's unlikely, though conceivable, that I missed something that should affect my ratings.

---

> ### Author Rebuttal · Authors · 2023-08-29
>
> Dear Reviewer 1ZMP:
>
> We sincerely thank you for the time and effort you have dedicated to reviewing our paper. Your valuable comments and constructive suggestions have been instrumental in improving the quality of our work. Below, we reply to the comments raised by the reviewer point-by-point.
>
> **Q1: Consider adding more evaluation metrics and methods.**
>
> A1: Thanks for your suggestion, we use the BLEU metric in the paper, showing the effectiveness of the method. In order to overcome the shortcomings of a single automatic evaluation metrics (BLEU score), we will use more diverse evaluation metrics to evaluate the translation quality of the model, such as chrf, COMET or GPT-4 etc.
>
> **Q2: More analysis about translation quality consistency and coherence.**
>
> A2: Automatically evaluating the consistency and coherence of document translations is difficult. On the one hand, we use BLEU score to evaluate it and show improvement in Table 4. On the other hand, we also use the method of manual evaluation. Specifically, we manually evaluated 100 pieces of test data, and about 30% of the samples showed improvement. We will follow your suggestions, which will be added to the appendix in the final version.
>
> **Q3: Quantitative analysis results for the second training strategy.**
>
> A3: Thank you for your suggestion. In our experiment, in order to enhance the robustness of the model, we adopted the second strategy. The manual evaluation demonstrated improvements, and we conducted case studies, but we apologize for not including the corresponding statistical analysis process in the article. We adopt your suggestion and give a quantitative analysis experiment, where we introduce some noise in the terminology constrained translation task. The results is (Aviation: 41.38 vs. 43.66;  Law: 52.45 vs. 54.98;  Entertainment: 29.73 vs. 31.50;). The experimental results show that the model with the second training strategy is more robust. We will include corresponding analysis in future version for better understanding by readers. Thanks again for your suggestion.
>
>
> **Q4: More details about test data.**
>
> A4: In the translation memory task and domain adaptation task, we use the public dataset UM-Corpus, and in the document translation task, we use IWSLT2015 as the test dataset. In the terminology constrained translation task, we were unable to find a suitable public test dataset. Therefore, based on a high-quality term dictionary, we extract the term information contained in each sentence to construct a test set and evaluate the performance of the model with it. Specifically, we use a 30w collated terminology dictionary. Thanks for your suggestion, we include more details in the future.
>
> **Q5: The difference in performance between the bloom-7b1 model and the model after finetuning.**
>
> We evaluated the origin bloom-7b1 model, and the results are not good. The BLEU score is basically below 5, and it is prone to translation off-target and repeated generation problems. The [R1] paper also reached the same conclusion. Typically, larger models have better translation results, however even a large model of 176B is still weaker than supervised training translation models (R[2]). Thank you for your suggestion, we will add the corresponding experimental results in the appendix later.\
> \
> \
> Thanking you again for your generosity with your valuable time invested in improving our paper.
>
> Yours sincerely,
>
> All author.\
> \
> \
> **References**
>
> [R1] Bawden R, Yvon F. Investigating the translation performance of a large multilingual language model: the case of bloom[J]. arXiv preprint arXiv:2303.01911, 2023.
>
> [R2] Jiao W, Wang W, Huang J, et al. Is ChatGPT a good translator? A preliminary study[J]. arXiv preprint arXiv:2301.08745, 2023.

---

### Official Review · Reviewer_afzn · 2023-08-15

**Soundness:** 4

**Excitement:**

4: Strong: This paper deepens the understanding of some phenomenon or lowers the barriers to an existing research direction.

**Missing References:**

BM25 algorithm

**Paper Topic And Main Contributions:**

In this paper, the authors delve into the utilization of pretrained language models for machine translation tasks. They introduce the MT2 model, which is a unified model capable of handling various machine translation tasks: translation memory and domain adaptation, document-level machine translation, and terminology-constrained translation. The paper exhibits strong motivation, a clear structure, and well-crafted writing. Furthermore, it is accompanied by a robust analysis and compelling experimental evidence.

**Questions For The Authors:**

Please see the above section

**Reasons To Accept:**

The paper is thoughtfully organized and skillfully composed, demonstrating a clear and logical structure that aids in understanding the content. The author's analysis is robust, contributing to the paper's credibility by offering thorough insights and well-founded conclusions. Moreover, the paper effectively illustrates the applicability of pretrained language models (LLMs) across multiple tasks in machine translation (MT) and highlights the concept of translation via in-context learning.


**Reasons To Reject:**

The abstract commences by discussing MT2's exploration of a singular model engaged in various tasks, particularly emphasizing its advantage in "knowledge transfer of different tasks" compared to using distinct models for each task. However, the evaluation illustrating MT2's benefits from other tasks, such as "knowledge transfer of different tasks," lacks clarity and substantiation. Despite this, the research effectively addresses other pertinent inquiries, displaying solid analysis.

The method of alignment mentioned in the abstract remains unclear within the paper, lacking explicit details on its application in the research. For example, its implementation in context retrieval should be explicitly detailed, especially in relation to its contribution to the in-context learning strategy employed in MT2. This clarification is essential to enhance the understanding of readers regarding the practicality of this alignment method.

**Reproducibility:**

4: Could mostly reproduce the results, but there may be some variation because of sample variance or minor variations in their interpretation of the protocol or method.

**Reviewer Confidence:**

4: Quite sure. I tried to check the important points carefully. It's unlikely, though conceivable, that I missed something that should affect my ratings.

---

> ### Author Rebuttal · Authors · 2023-08-29
>
> Dear Reviewer afzn:
>
>   We sincerely thank you for the time and effort you have dedicated to reviewing our paper. Your valuable comments and constructive suggestions have been instrumental in improving the quality of our work. Below, we reply to the comments raised by the reviewer point-by-point.
>
> **Q1: Knowledge transfer of different tasks lacks clarity and substantiation.**
>
> A1: Previous work demonstrates that multi-task learning benefits the learning of machine translation models, such as [R1] and [R2]. In the paper [R3], single-sentence and multi-sentence joint training has also improved the performance of the model. We followed the successful experience and ideas of previous work, as this is not our main contribution point, we did not analyze it in detail. Thank you very much for your suggestion, we will add corresponding statements in the Related Work section to better enable readers to understand the relevant content.
>
> **Q2: The clarification of the mentioned alignment method is essential to enhance the understanding of readers regarding the practicality of this alignment method.**
>
> A1: One of its contributions is that it can make up for the lack of data in term translation tasks. This alignment-based method can construct large-scale context-enhanced training data, which is conducive to the training of ICL strategies and improves the in-context learning ability of the model. In addition, the data can be generalized not only at the term level but into phrase-level constraints, which helps to improve the generalization of the model. Thanks for your suggestion, we’ll add more detailed descriptions in the later version.
>
> **Q3: Missing reference for BM25 algorithm.**
>
> A3: Thanks for pointing this out, this is our negligence, we will add the reference of BM25 algorithm (R[4]).\
> \
> Thanking you again for your generosity with your valuable time invested in improving our paper.
>
> Yours sincerely,
>
> All author.\
> \
> \
> **References**
>
> [R1] Zhou S, Zeng X, Zhou Y, et al. Improving robustness of neural machine translation with multi-task learning[C]//Proceedings of the Fourth Conference on Machine Translation (Volume 2: Shared Task Papers, Day 1). 2019: 565-571.
>
> [R2] Luong M T, Le Q V, Sutskever I, et al. Multi-task sequence to sequence learning[J]. arXiv preprint arXiv:1511.06114, 2015.
>
> [R3] Sun Z, Wang M, Zhou H, et al. Rethinking document-level neural machine translation[J]. arXiv preprint arXiv:2010.08961, 2020.
>
> [R4] Robertson S, Zaragoza H. The probabilistic relevance framework: BM25 and beyond[J]. Foundations and Trends® in Information Retrieval, 2009, 3(4): 333-389.

---

### Meta-Review · Area_Chair_tLAM · 2023-09-15

**Recommendation:** 5

**Metareview:**

This paper proposes a multi-task machine translation model called MT2 and uses in-context learning (ICL) paradigm for model training. It has a clear structure, extensive experiments and analyses.
Authors demonstrate the effectiveness of large language models (LLMs) for various machine translation tasks via in-context learning with convincing results. Some minor aspects could be improved for the camera-ready: references (see reviewers comments), use more diverse evaluation metrics of translation quality (not only BLEU), inclusion of detailed results for the second training strategy, minor improvements in grammar and style.

---

### Decision · Program_Chairs · 2023-10-07

**Decision:**

Accept-Main

**Comment:**

This paper proposes a multi-task machine translation model called MT2 and uses in-context learning (ICL) paradigm for model training. It has a clear structure, extensive experiments and analyses.
Authors demonstrate the effectiveness of large language models (LLMs) for various machine translation tasks via in-context learning with convincing results. Some minor aspects could be improved for the camera-ready: references (see reviewers comments), use more diverse evaluation metrics of translation quality (not only BLEU), inclusion of detailed results for the second training strategy, minor improvements in grammar and style.